# Attack-Agnostic Adversarial Detection

**Jiaxin Cheng**[1,3]    **Mohamed Hussein**[1,3,4]    **Jay Billa**[1,3]    **Wael AbdAlmageed**[1,2,3]

[1]University of Southern California, Information Sciences Institution
[2]USC Ming Hsieh Department of Electrical and Computer Engineering
[3]Visual Intelligence and Multimedia Analytics Laboratory
[4]Faculty of Engineering, Alexandria University, Egypt
{chengjia,mehussein,jbilla,wamageed}@isi.edu

## Abstract

The growing number of adversarial attacks in recent years gives attackers an advantage over defenders, as defenders must train detectors after knowing the types of attacks, and many models need to be maintained to ensure good performance in detecting any upcoming attacks. We propose a way to end the tug-of-war between attackers and defenders by treating adversarial attack detection as an anomaly detection problem so that the detector is agnostic to the attack. We quantify the statistical deviation caused by adversarial perturbations in two aspects. The Least Significant Component Feature (LSCF) quantifies the deviation of adversarial examples from the statistics of benign samples, and Hessian Feature (HF) reflects how adversarial examples distort the landscape of the models' optima by measuring the local loss curvature. Empirical results show that our method can achieve an overall ROC AUC of 94.9%, 89.7%, and 97.9% on CIFAR10, CIFAR100, and SVHN, respectively. Code available https://github.com/cplusx/HEAD

## 1  Introduction

Despite the success of deep neural networks (DNNs) in computer vision [23, 46, 17], natural language processing [18, 51, 11] and speech recognition [9, 56], DNNs are notoriously vulnerable to adversarial attacks[48] that inject carefully crafted imperceptible perturbations into the input and are able to deceive the model with a great chance of success.

There are three main orthogonal approaches for combating adversarial attacks — (i) Using adversarial attacks as a data augmentation mechanism by including adversarially perturbed samples in the training data to induce robustness in the trained model [30, 57, 58, 59, 5, 55, 3, 54]; (ii) Preprocessing the input data with a denoising function or deep network [27, 8, 13, 15, 34] to counteract the effect of adversarial perturbations; and (iii) Training an auxiliary network to *detect* adversarial examples and deny providing inference on adversarial samples [35, 26, 42, 43, 14, 28, 29, 25, 6, 16, 10, 1]. Our work falls under adversarial example detection as it does not require retraining the main network (as in adversarial training) nor degrade the input quality (as in preprocessing defenses).

Existing adversarial example detection methods [28, 29, 25, 6, 16] need to train auxiliary networks in a binary classification manner (*e.g.* benign versus adversarial attack(s)). The shortcoming of this strategy is that the detector is trained on specific attack(s) that are available and known at training time. To ensure good detection performance at inference time, the detection network needs to be trained on a large number of attacks. Otherwise, the detection network will perform poorly on attacks unseen during training (*i.e.* out of domain) or even on attacks seen during training due to overfitting [40]. We argue that a good adversarial detection method should be able to detect any adversarial attack, even if the defender is unaware of the type of adversarial attack. To this end, we propose to frame the adversarial sample detection as an anomaly detection problem, in which only one detection model is constructed and trained on only benign samples, such that the detection model is *attack-agnostic*.

2022 Trustworthy and Socially Responsible Machine Learning (TSRML 2022) co-located with NeurIPS 2022.

We propose an anomaly detection framework for identifying adversarial examples by measuring the statistical deviations caused by adversarial perturbations. We consider the deviation of two complementary features that reflect the interaction of adversarial perturbation with the data and models. The first feature is Least Significant Component Feature (LSCF), which maps data to a subspace where the distribution of benign images is compact, while the distribution of adversarial images is spread. The second feature is Hessian Feature (HF), which uses the second order derivatives as a measure of the distortion caused by adversarial perturbation to the model's loss landscape. Our results underscore the utility of each of the two features and their complementary nature.

The contributions of this paper are:

1. An anomaly detection framework for adversarial detection that measures statistical deviation caused by adversarial perturbations on two proposed features, LSCF and HF, which are theoretically justified and capture the interaction of adversarial perturbations with data and model.
2. Empirical analysis demonstrating the effectiveness of our method on detecting eight different adversarial attacks on three datasets, achieving AUC of up to 94.9%.
3. Comprehensive evaluation of our method analyzing its computational efficiency, sensitivity to hyper parameters, cross-model generalization, and closeness to binary classification upper bound.

## 2 Attack-Agnostic Adversarial Detection

### 2.1 Challenges And Rationale

A fundamental assumption of existing adversarial attack detection [35, 26, 42, 43, 14, 28, 29, 25, 6, 16, 10, 1] as well as adversarial augmentation methods [28, 29, 25, 6, 16] is that adversarial attacks are known and samples can easily be generated using these attacks to train the detector or augment the main model being defended. This assumption, however, is not realistic, since more often than not the defender does not know the attacks *a priori* and therefore samples cannot be easily generated to train a supervised adversarially-augmented detector. This motivates us to frame the adversarial detection as an anomaly detection problem. More formally, the task of the defender is to protect the model trained on *only benign examples* $X$ against adversarial examples $\widehat{X}$ that are unknown during training. The trained anomaly detector $D$ will give a score $s(x) = D(f(\mathbf{x}))$ for each testing sample $\mathbf{x}$, indicating the likelihood that $\mathbf{x}$ is an adversarial attack, where $f \in \mathbb{R}^m \times \mathbb{R}^n$ is a feature extraction function and $m$ and $n$ are the dimensions of input and feature spaces, respectively. For $f$, we propose two complementary features that

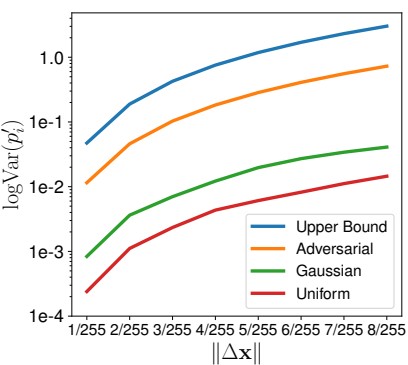

Figure 1: Upperbound in eq. (2) and deviations of $p_i$ caused by FGSM, Gaussian and Uniform perturbation, average over 1000 CIFAR10 images.

reflect the interaction between adversarial perturbation and dataset (Least Significant Component Feature) as well as DNN models (Hessian Feature). We call our features HEAD, which stands for Hessian and Eigen-decomposition-based Adversarial Detection.

### 2.2 Least Significant Component Feature (LSCF)

We suspect that global image context features will not work well for detecting adversarial attacks, since they tend to miss subtleties introduced by the adversarial perturbations. Therefore, we extract LSCF that is sensitive to small imperceptible image noise. We use principal component analysis (PCA) to project the raw benign images to a space with orthonomal basis (*i.e.* eigenvectors) in which different dimensions are linearly uncorrelated. Rather than retaining the projections that correspond to the largest eignvalues (*i.e.* eigenfaces [50]), we retain the projections on the directions with smallest eignvalues. Hence, the features are the least significant components of the data.

Formally, let the training data $\mathbf{D} \in \mathbb{R}^{N \times m}$ be of $N$ samples and $m$ input dimensions. Its covariance matrix $\mathbf{C} = \mathbf{D}^\top \mathbf{D}/(N-1)$ can be decomposed into $\mathbf{C} = \mathbf{V}\mathbf{L}\mathbf{V}^\top$, where the columns of $\mathbf{V} = [\mathbf{v}_1\mathbf{v}_2...\mathbf{v}_m]$ are the eigenvetors of $\mathbf{C}$, and $\mathbf{L}$ is a diagonal matrix having eigenvalues of $\mathbf{C}$ in

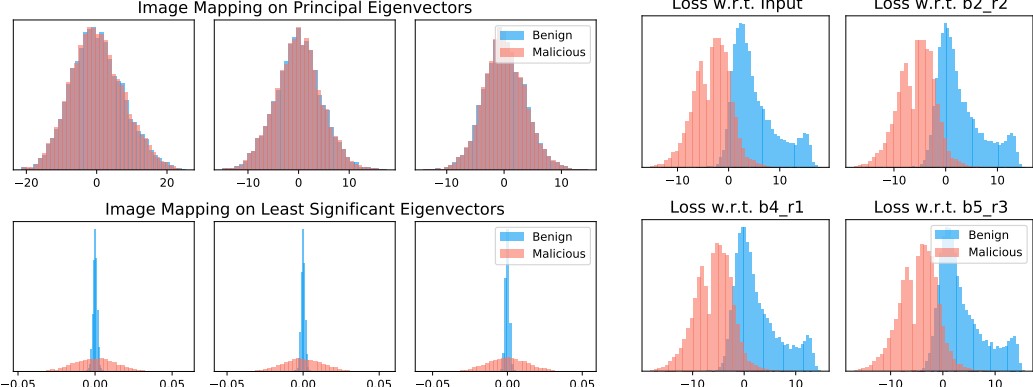

Figure 2: Distribution of images' mapping on three eigenvectors of Principal Components (upper) and Least Significant Components (bottom) with PGD10 ($\epsilon$=8/255) on CIFAR10.

Figure 3: The Hessian modulus distribution of benign and PGD10 ($\epsilon$=8/255) CIFAR10. $bx\_ry$ is block $x$ ReLU $y$ in VGG16.

descending order on its diagonal (*i.e.*, $\mathrm{diag}(\mathbf{L}) = [\lambda_1 \lambda_2 ... \lambda_m], \lambda_i \geq \lambda_{i+1} \forall i \in \{1..m-1\}$). The LSCF of image $\mathbf{x}$ is transformed by the eigenvectors $\mathbf{v}$ that have the smallest eigenvalues.

$$f_{LSCF}(\mathbf{x}) = \mathbf{x}^\top \mathbf{v} \in \mathbb{R}^{1 \times d} \tag{1}$$

where $d$ is the dimension of LSCF and $\mathbf{v} = [\mathbf{v}_m \mathbf{v}_{m-1} ... \mathbf{v}_{m-d+1}] \in \mathbb{R}^{m \times d}$ is the last $d$ columns of $\mathbf{V}$. We explain the reason for mapping images on the least significant eigenvectors by estimating an upper bound on the expected deviation caused by perturbation for different eigenvectors. Let $p_i = \mathbf{x}^\top \mathbf{v}_i$ be the mapping of image $\mathbf{x}$, and $p_i' = (\mathbf{x} + \Delta\mathbf{x})^\top \mathbf{v}_i$ be the mapping of the adversarially perturbed image on the $i$th eigenvector, respectively. Since the transformation is linear, we have $\Delta p_i = \Delta\mathbf{x}^\top \mathbf{v}_i$. The variance of $p_i$ measures the spread of data in $\mathbf{v}_i$'s direction and hence has $\mathrm{Var}(p_i) = \lambda_i$, while the variance of $p_i'$ has an upper bound in eq. (2) (See proof in the Appendix A).

$$\mathrm{Var}(p_i') \leq \lambda_i + \|\Delta\mathbf{x}\|^2 \tag{2}$$

The empirical analysis in Figure 1 suggests that the actual variance of projected perturbed images on the least significant eigenvector is much closer to that upper bound than random noises.

When the difference between $\mathrm{Var}(p_i')$ and $\mathrm{Var}(p_i)$ is large, we can easily distinguish benign images from adversarial images by mapping them onto eigenvector $\mathbf{v}_i$. We quantify this difference by the ratio of $\mathrm{Var}(p_i')/\mathrm{Var}(p_i)$, which has an upper bound of $1 + \|\Delta\mathbf{x}\|^2/\lambda_i$. Since the perturbation budget $\|\Delta\mathbf{x}\|$ is predefined, the value of $\lambda_i$ determines the differentiability between $p_i'$ and $p_i$. The smaller the value of $\lambda_i$, the easier to distinguish adversarial from benign images. As a result, mapping data onto the least significant components gives highest the distinguishability. Figure 2 visualizes the distributions of projected values for 1,000 adversarial and benign images on major and least significant principal components. We can see that the distributions are indistinguishable in the major PCA components, but are clearly distinguishable in the least significant components.

## 2.3 Hessian Feature

When studying model optimization, [21, 60] observed that perturbation on model weights can improve generalization, and [53, 49, 52] later proved that such improvement happens because the perturbation changes the smoothness of the loss function's landscape, which can be measured by the Hessian matrix of the loss. Motivated by this observation, we hypothesize that the Hessian can be used to characterize the loss landscape and find locations that are exploited by the adversarial perturbations. In fact, the adversarial attack creation problem is very similar to the problem of model optimization in the sense that they bear similarity to Lagrange duality.

More formally, model optimization can be expressed by Equation (3)

$$\underset{\mathbf{W}}{\text{minimize}} \quad L[Y, f(\mathbf{W}, \mathbf{X})] \qquad \text{s.t.} \quad \mathbf{X} = \mathbf{D} \tag{3}$$

while the target of adversarial attack can be written as Equation (4)

$$\underset{\mathbf{X}}{\text{maximize}} \quad L[Y, f(\mathbf{W}^*, \mathbf{X})] + \sum_{u_i} u_i \|\mathbf{X}_i - \mathbf{D}_i\|_p \tag{4}$$

$$\text{s.t.} \quad u_i \leq 0, \quad \forall i \in [1, N]$$

where $L$ is the loss function, $\mathbf{W}$ is the parameters of the model $f$, $\mathbf{D}$ is the training data, $Y$ is the target, and $\mathbf{W}^*$ is the optimized (*i.e.* trained) model weights that achieves $\inf_{\mathbf{W}}(L[Y, f(\mathbf{W}, \mathbf{X})])$.

If we regard the Lagrange regularizers as the adversarial perturbation constraints (*i.e.*, $l_\infty$, $l_2$ or $l_0$), Equation (4) can be seen as the adversarial attack against the dataset where $L[Y, f(\mathbf{W}^*, \mathbf{X})]$ corresponds to maximizing the prediction error and $\sum_{u_i} u_i \|\mathbf{X}_i - \mathbf{D}_i\|_p$ corresponds to limiting the perturbation budget under $l_p$ constraint.[1] Such correspondence in duality motivates us to measure the statistical deviation of the Hessian matrix to detect adversarial examples. The Hessian we use is the second-order derivative of the loss *w.r.t.* the input or the outputs of the intermediate layers, *i.e.*

$$\mathrm{H} \equiv \frac{\partial^2 L(\mathbf{x})}{\partial^2 \mathbf{x}} \quad \text{where} \quad \mathrm{H}[i, j] = \frac{\partial^2 L(\mathbf{x})}{\partial x_i \partial x_j}, \tag{5}$$

$\mathbf{x}$ is input or the outputs of the intermediate layers of the model (*e.g.*, outputs of ReLU layers), $x_i$ and $x_j$ are the $i$th and $j$th entry (*e.g.*, pixels in image) of the input, respectively. Since the size Hessian matrix is proportional to the square of the input dimension, we use the $l_1$ modulus of the Hessian as an approximation of the Hessian to avoid computational problems for anomaly detection models due to the curse of high dimensionality [2]. Please note that the ground truth label is not required during computation and any choice of label will give the same result since Hessian only show the curvature of the loss landscape. Figure 3 shows the Hessian distribution of benign and PGD10 images. The distributions suggest that the modulus of the Hessian can be used to separate benign and adversarial samples. Nevertheless, our final Hessian feature includes multiple dimensions by using the moduli of Hessian matrices for multiple network layers along with the Hessian matrix for the input.

## 3 Experimental Evaluation

### 3.1 Benchmarks and Baselines

We evaluate the performance of HEAD on CIFAR10 [22], CIFAR100 [22] (50,000 training / 10,000 testing), and SVHN [38] (73,257 training / 26,032 testing) datasets. Our HEAD-based anomaly detectors are trained on benign images only. We base our experiments on the VGG16 [46] model.

**Baseline features:** We compare HEAD against one naive image feature (PCA), two hand-crafted features (LID [29] and Mahalanobis [25]), and one learned deep feature (DSVDD [44]). We use 32 principal components for PCA. We choose LID and Mahalanobis as they do not require supervision to compute features and hence are suitable for anomaly detection, and we follow the original papers but change the target network to VGG16[46] to provide a fair comparison to the HEAD features. DSVDD integrates both the feature extractor and anomaly detector. We train the feature extractor for 100 epochs and tune the anomaly detector for 50 epochs, following [44].

**HEAD features:** We extract a 32-dimensional LSCF feature. We compute Hessian of the loss *w.r.t.* the input and the intermediate features from the ReLU layers to form a 13-dimensional Hessian feature. The LSCF and HF are concatenated to a 45-dimensional HEAD feature for each image.

**Anomaly Detectors:** We train both kernel density estimator (KDE) and One-Class SVM (OCSVM) based anomaly detectors on each set of features. For KDE, we evaluate using Gaussian, Epanechnikov, exponential, linear, and uniform kernels. For OCSVM, we evaluate using RBF, Sigmoid, linear and polynomial kernels. We also conduct a grid search for hyperparameters and report the best performance, the corresponding ablation studies can be found in the Appendix D.1.

**Adversarial Attacks:** Each anomaly detector is evaluated across eight standard attacks. For $l_\infty$ with max perturbation 8/255, we use (1) Fast Gradient Sign Method (FGSM) [12], (2) Projected Gradient Descent (PGD10) [31] and (3) Basic Iterative Method (BIM) [24], both with 10 iterations. For $l_2$ attacks with total perturbation budget of 1, we use (4) DeepFool [37] and (5) Carlini & Wagner [4]. For $l_0$, we use (6) OnePixel [47] and (7) SparseFool [36] with hyperparameter $lam = 3$. For combined attacks with perturbation budget of 8/255 under $l_\infty$, we use (8) AutoAttack [7].

---

[1]We slightly abuse the name of Lagrange regularizer as the norm $\|\cdot\|_p$ is not required in Lagrange duality.

| Dataset | Method | $l_\infty$ Attacks | | | $l_2$ Attacks | | $l_0$ Attacks | | Combined | Overall |
|---|---|---|---|---|---|---|---|---|---|---|
| | | PGD10 | FGSM | BIM | DeepFool | CW | SparseFool | OnePixel | AutoAttack | |
| CIFAR10 | PCA+OCSVM | 0.497 | 0.498 | 0.497 | 0.500 | 0.500 | 0.109 | 0.497 | 0.293 | 0.424 |
| | PCA+KDE | 0.501 | 0.498 | 0.500 | 0.501 | 0.500 | 0.502 | 0.500 | 0.501 | 0.500 |
| | DSVDD [44] | 0.569 | 0.614 | 0.566 | 0.505 | 0.507 | 0.901 | 0.505 | 0.958 | 0.641 |
| | LID+OCSVM | 0.551 | 0.596 | 0.575 | 0.583 | 0.585 | 0.914 | 0.559 | 0.968 | 0.666 |
| | LID [29]+KDE | 0.610 | 0.702 | 0.639 | 0.654 | 0.656 | 0.924 | 0.615 | 0.971 | 0.721 |
| | Mah.+OCSVM | 0.880 | 0.787 | 0.898 | **0.852** | 0.837 | 0.963 | 0.668 | **0.989** | 0.859 |
| | Mah. [25]+KDE | 0.896 | 0.887 | 0.893 | 0.603 | 0.587 | 0.899 | 0.578 | 0.966 | 0.789 |
| | HEAD+OCSVM (Ours) | 0.999 | **0.999** | 0.999 | 0.841 | 0.941 | 0.985 | 0.821 | 0.988 | 0.947 |
| | HEAD+KDE (Ours) | **1.000** | **0.999** | **1.000** | 0.846 | **0.943** | **0.986** | **0.825** | **0.989** | **0.949** |
| CIFAR100 | PCA+OCSVM | 0.497 | 0.497 | 0.497 | 0.500 | 0.500 | 0.221 | 0.497 | 0.353 | 0.445 |
| | PCA+KDE | 0.498 | 0.501 | 0.499 | 0.500 | 0.500 | 0.501 | 0.500 | 0.497 | 0.500 |
| | DSVDD | 0.568 | 0.629 | 0.564 | 0.502 | 0.504 | 0.777 | 0.501 | 0.852 | 0.612 |
| | LID+OCSVM | 0.570 | 0.579 | 0.581 | 0.504 | 0.501 | 0.758 | 0.520 | 0.845 | 0.607 |
| | LID+KDE | 0.642 | 0.655 | 0.654 | 0.511 | 0.515 | 0.768 | 0.549 | 0.849 | 0.643 |
| | Mah.+OCSVM | 0.708 | 0.719 | 0.709 | **0.816** | 0.811 | 0.772 | **0.883** | **0.916** | 0.792 |
| | Mah.+KDE | 0.845 | 0.926 | 0.848 | 0.535 | 0.541 | 0.760 | 0.530 | 0.798 | 0.723 |
| | HEAD+OCSVM (Ours) | **0.999** | 0.999 | **0.998** | 0.728 | 0.814 | 0.898 | 0.819 | 0.906 | 0.895 |
| | HEAD+KDE (Ours) | **0.999** | **1.000** | **0.998** | 0.733 | **0.816** | **0.901** | 0.820 | 0.907 | **0.897** |
| SVHN | PCA+OCSVM | 0.499 | 0.501 | 0.499 | 0.500 | 0.500 | 0.242 | 0.497 | 0.342 | 0.448 |
| | PCA+KDE | 0.500 | 0.499 | 0.499 | 0.499 | 0.499 | 0.501 | 0.500 | 0.495 | 0.499 |
| | DSVDD | 0.717 | 0.812 | 0.714 | 0.524 | 0.527 | 0.911 | 0.521 | 0.981 | 0.713 |
| | LID+OCSVM | 0.680 | 0.640 | 0.693 | 0.654 | 0.680 | 0.927 | 0.525 | 0.984 | 0.723 |
| | LID+KDE | 0.761 | 0.747 | 0.772 | 0.726 | 0.749 | 0.938 | 0.560 | 0.986 | 0.780 |
| | Mah.+OCSVM | 0.747 | 0.699 | 0.766 | **0.917** | 0.941 | 0.966 | 0.663 | **0.994** | 0.837 |
| | Mah.+KDE | 0.833 | 0.748 | 0.848 | 0.904 | 0.914 | 0.909 | 0.638 | 0.971 | 0.846 |
| | HEAD+OCSVM (Ours) | **1.000** | **1.000** | **1.000** | 0.868 | 0.975 | 0.992 | **0.954** | **0.994** | 0.973 |
| | HEAD+KDE (Ours) | **1.000** | **1.000** | **1.000** | **0.917** | **0.979** | **0.994** | 0.946 | **0.994** | **0.979** |

Table 1: The ROC AUC performance on detecting eight adversarial attacks. Best performance is reported in **bold** and second best with underline.

## 3.2 Evaluation Results

Each anomaly detector is evaluated using the area under receiver operating characteristic curve (ROC AUC) on all adversarial attacks. The results are summarized in Table 1. Note that the same anomaly detector is used to detect **any** of the eight attacks.

We observe that, with very few exceptions, across all anomaly detection variants, HEAD-based anomaly detectors demonstrate the best performance. In general, features that represent holistic image features, such as PCA and DSVDD, do not perform well. The subtle and localized adversarial perturbations are likely overlooked by these global image features. HEAD features, in particular, perform well against both $l_\infty$ attacks and AutoAttack. We find that AutoAttack is easy to detect for all but the PCA-based anomaly detectors. We speculate that the reason for this behavior is that ensemble attacks leave more traces of tampering and are therefore easier to detect. HEAD features appear to be particularly robust to $l_\infty$ attacks vis-á-vis the other approaches. Even on $l_2$ and $l_0$ attacks, HEAD features perform better than most of the compared features. Across all attacks, HEAD features achieve almost 95% AUC on CIFAR10 and SVHN, and almost 90% AUC on CIFAR100.

## 3.3 Cross-model Adversarial Detection

Adversarial examples generated by one model are known to be transferrable in that they can deceive a trained model with a different architecture [39]. We validate HEAD's ability to detect cross-model attacks by generating adversarial images with a ResNet18 model and detecting malicious images with a VGG16 model. For cross-model adversarial detection with LID [29] and Mahalanobis [25], we find that the baseline anomaly detectors perform quite poorly. To provide a stronger comparison, we instead compare against the LID and Mahalanobis supervised models. (Note that supervised models are trained on adversarial images of VGG16 but evaluated on adversarial images of ResNet18.) The training setting for supervised model can be found in Appendix C. For HEAD however, we use the same anomaly detection models as in Section 3.1. As shown in Table 2, across all datasets and attacks, HEAD based anomaly detectors significantly outperform the supervised LID and Mahalanobis feature based models. Only on $l_2$ DeepFool attack, Mahlanobis-based supervised model slightly outperforms the HEAD-based anomaly detector.

| Dataset | Method | $l_\infty$ Attacks | | | $l_2$ Attacks | | $l_0$ Attacks | | Combined | Overall |
|---|---|---|---|---|---|---|---|---|---|---|
| | | PGD10 | FGSM | BIM | DeepFool | CW | SparseFool | OnePixel | AutoAttack | |
| CIFAR10 | LID (Binary Classification) | 0.594 | 0.901 | 0.588 | 0.617 | 0.652 | 0.830 | 0.677 | 0.876 | 0.717 |
| | Mah. (Binary Classification) | 0.702 | 0.991 | 0.704 | **0.628** | 0.658 | 0.838 | 0.674 | 0.740 | 0.742 |
| | HEAD+OCSVM (Ours) | **1.000** | **1.000** | **0.999** | 0.589 | 0.880 | 0.969 | **0.882** | **0.988** | 0.913 |
| | HEAD+KDE (Ours) | **1.000** | **1.000** | **0.999** | 0.590 | **0.883** | **0.970** | 0.881 | **0.988** | **0.914** |
| CIFAR100 | LID (Binary Classification) | 0.650 | 0.831 | 0.636 | 0.499 | 0.504 | 0.810 | 0.596 | 0.842 | 0.671 |
| | Mah. (Binary Classification) | 0.737 | 0.985 | 0.713 | **0.531** | 0.556 | 0.839 | 0.662 | 0.752 | 0.722 |
| | HEAD+OCSVM (Ours) | **0.999** | **0.999** | **0.998** | 0.527 | 0.762 | 0.901 | 0.814 | 0.917 | 0.861 |
| | HEAD+KDE (Ours) | **0.999** | **0.999** | **0.998** | 0.530 | **0.765** | **0.905** | **0.815** | **0.919** | **0.866** |
| SVHN | LID (Binary Classification) | 0.776 | 0.777 | 0.788 | 0.593 | 0.609 | 0.931 | 0.583 | 0.890 | 0.743 |
| | Mah. (Binary Classification) | 0.797 | 0.847 | 0.808 | **0.647** | 0.659 | 0.942 | 0.651 | 0.801 | 0.769 |
| | HEAD+OCSVM (Ours) | **1.000** | **1.000** | **1.000** | 0.605 | 0.898 | **0.993** | 0.928 | 0.994 | 0.927 |
| | HEAD+KDE (Ours) | **1.000** | **1.000** | **1.000** | 0.602 | **0.901** | 0.992 | **0.930** | **0.995** | **0.928** |

Table 2: The ROC AUC performance on detecting cross model adversarial attacks. Best performance is reported in **bold** and second best with underline.

## 3.4 Sensitivity and Ablation Studies

To further understand the properties of the HEAD features, we conduct experiments on CIFAR10 to evaluate (i) effectiveness of LSCF and HF, (ii) performance gap between anomaly detection and binary classification, (iii) method sensitivity to the anomaly detectors, and (iv) method robustness when distinguish benign noisy images and adversarial images. The result of (ii), (iii) and (iv) are provided in the appendix D due to page limitation.

**Effectiveness of LSCF And HF components of HEAD:** To compare the effectiveness of LSCF and HF we ablate on the number of feature components. Specifically, for LSCF, we use 0, 4, 16, 32, 64-dimensional feature variants. For HF, we use 0, 1 (only input), 5 (from input to $b2\_r2$), 9 (from input to $b4\_r1$) and 13-dimensional (from input to $b5\_r3$) features. When one feature size (LSCF or HF) is changed, we use the best number of feature components for the other feature. Results are detailed in the Table 3. Both features show improved performance as the number of feature components increases. We observe that LSCF and HF are complementary in that the largest performance gains are obtained when LSCF and HF are concatenated. For LSCF, performance plateaus at 32 dimensions. Based on this ablation study, we choose 13-dimensional HF and 32-dimensional LSCF in the experiments for the remainder of the paper.

| HF Dimension | ROC AUC | Improve |
|---|---|---|
| 0 | 0.885 | - |
| 1 | 0.936 | +0.051 |
| 5 | 0.946 | +0.010 |
| 9 | 0.948 | +0.002 |
| 13 | 0.949 | +0.001 |

| LSCF Dimension | ROC AUC | Improve |
|---|---|---|
| 0 | 0.860 | - |
| 4 | 0.923 | +0.063 |
| 16 | 0.939 | +0.016 |
| 32 | 0.949 | +0.010 |
| 64 | 0.949 | +0.000 |

Table 3: The effectiveness of different dimensional Hessian Feature (left) and Least Significant Component Feature (right). The performance is shown in ROC AUC over all attacks. Dimension=0 implies the feature is not used. The right column shows the incremental performance improvement over the prior row.

## 4 Conclusion

We frame adversarial detection as an anomaly detection problem to better reflect the challenge of detecting adversarial examples in real life. We propose Hessian and Eigen-decomposition-based Adversarial Detection, which measures the statistical deviation caused by adversarial perturbation on two complementary features: LSCF, which captures the deviation of adversarial images from the benign data, and HF, which reflects the deformation of the model's loss landscape at adversarialy perturbed images. We provide the theoretical rationale behind using LSCF and HF. Empirical results prove the effectiveness of HEAD and show that comparable performance to binary classification based adversarial detection can be achieved with anomaly detection. Our method does not use any outlier examples upon training anomaly detection, which could be a limitation in cases where outlier examples are easy to obtain. We defer the study of this case to our future research.

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

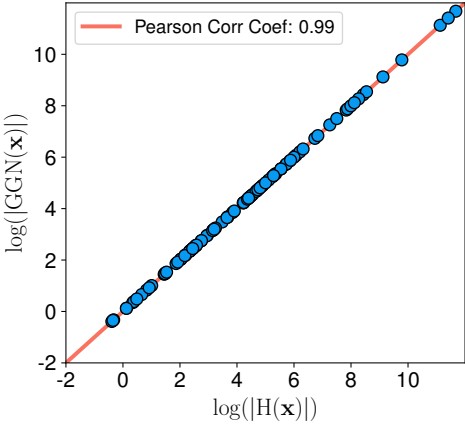
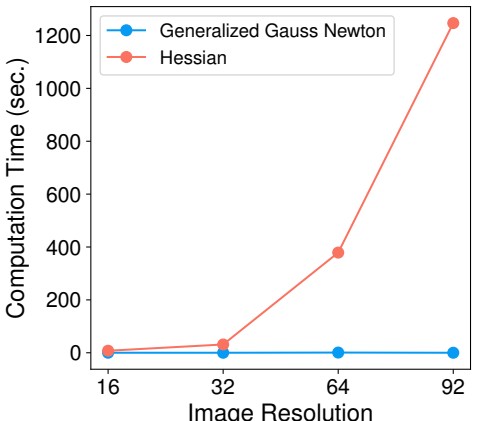

Figure 4: As a good approximation for Hessian, there is a strong correlation between the matrix modulus of GGN and Hessian.

Figure 5: The computation time of Hessian and GGN under different image sizes.

## A Proof of Equation (2)

We prove that the variance of $i$th eigenvector of adversarial image $\text{Var}(p_i')$ has an upper bound of $\lambda_i + \mathbf{E}(\Delta p_i^2)$. Assuming that the adversarial perturbation is independent from the data, we can have

$$\text{Var}(p_i') = \text{Var}(p_i) + \text{Var}(\Delta p_i) = \lambda_i + \mathbf{E}(\Delta p_i^2) - \mathbf{E}(\Delta p_i)^2 \leq \lambda_i + \mathbf{E}(\Delta p_i^2) \tag{6}$$

Further, the expected deviation of perturbation $\mathbf{E}(\Delta p_i^2)$ can be no larger than the norm of the perturbation $\|\Delta\mathbf{x}\|^2$ as shown in Equation (7)

$$\mathbf{E}(\Delta p_i^2) = \mathbf{E}((\Delta\mathbf{x}^\top\mathbf{v}_i)^2) \overset{①}{\leq} \mathbf{E}(\|\Delta\mathbf{x}\|^2\|\mathbf{v}_i\|^2) \overset{②}{=} \mathbf{E}(\|\Delta\mathbf{x}\|^2) \overset{③}{=} \|\Delta\mathbf{x}\|^2 \tag{7}$$

where ① is due to the Cauchy–Schwarz inequality, ② holds as $\mathbf{v}_i$ is an eigenvector with $\|\mathbf{v}_i\| = 1$, and ③ holds since for adversarial attacks, the injected perturbation budget $\|\Delta\mathbf{x}\|$ is the same for all images (if the maximum budget is always achieved). By combining Equations (6) and (7), we obtain that $\text{Var}(p_i')$ can be no larger than $\lambda_i + \|\Delta\mathbf{x}\|^2$.

## B Generalized Gauss-Newton Matrix for Approximating Hessian Matrix

We compute the Generalized Gauss-Newton matrix [45, 32, 33] instead to significantly speed up calculating the Hessian. Let $L$ be the loss, $\mathbf{x}$ be the variable to the loss (*e.g.* images) and $\mathbf{z}$ be the inputs of penultimate layer (*e.g.* the Softmax layer in DNNs). The GGN can be computed as

$$\mathrm{G} = (\mathrm{J}_\mathbf{x}^\mathbf{z})^\top \otimes \mathrm{H}_\mathbf{z}^L \otimes \mathrm{J}_\mathbf{x}^\mathbf{z} \tag{8}$$

where $\otimes$ is the matrix multiplication, $\mathrm{J}_\mathbf{x}^\mathbf{z}$ is the Jacobian of the penultimate layer $\mathbf{z}$ *w.r.t.* the input $\mathbf{x}$ and $\mathrm{H}_\mathbf{z}^L$ is the Hessian of loss $L$ *w.r.t.* penultimate layer $\mathbf{z}$. Please note that the ground truth label is not required during computation and any choice of label will give the same result since GGN/Hessian modulus only show the curvature of the loss landscape. Though GGN approximates Hessian well [45, 32, 33], it is unclear how good the modulus of GGN approximates the modulus of Hessian. We empirically show the approximation accuracy by randomly picking 1,000 samples from CIFAR10 and computing their Hessian and GGN. Figure 4 shows the matrix modulus of Hessian and GGN while Figure 5 summarizes the computation time of Hessian and GGN under different image sizes. We notice that GGN is strongly correlated to Hessian, while being much more computationally efficient to calculate. Therefore we use GNN as a substitute for Hessian [45].

## C  Training Setting For Supervised Binary Classifier in Section 3.3

The supervised model is a binary classifier consisting of four fully connected layers with output dimensions of 64, 32, 8, and 1. ReLU layers and batch normalization layers [20] are attached after the first three fully connected layers, and Sigmoid layer after the last one. We optimize this model with SGD [41], with learning rate = 0.001, for 100 epochs using binary cross-entropy loss.

## D  Additional Ablation Studies

### D.1  Sensitivity to Anomaly Detector Parameters

KDE requires a choice of kernel and bandwidth, and OCSVM requires a selection of kernel and $\nu$ value. We evaluate KDE using Gaussian, Epanechnikov, exponential, linear, and uniform kernels with bandwidth values from 1 to 25. Figure 6 shows the overall AUCs for these parameter values. The results indicate the choice of the kernel is not critical, since all kernels achieve similar performance with an appropriate bandwidth choice. For OCSVM, we evaluate using RBF, Sigmoid, linear and polynomial kernels with $\nu$ values from 0.1 to 0.9. Results are shown in Figure 7. Unlike KDE, OCSVM is sensitive to the choice of kernel, with the RBF kernel significantly outperforming all other kernels. That said, with an appropriate choice of hyperparameters, HEAD-based detector performance is insensitive to the choice of anomaly detector.

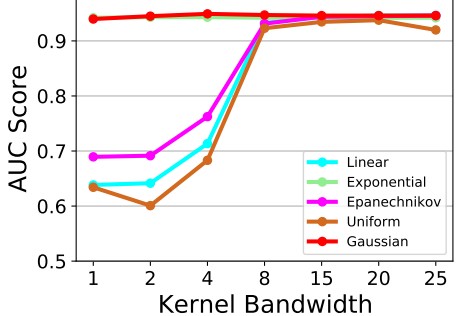
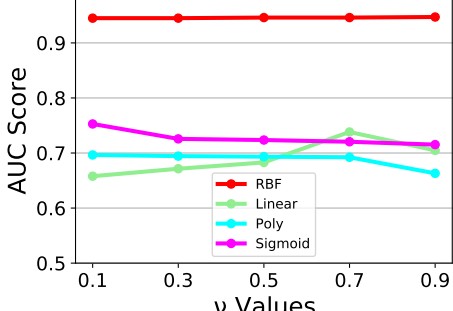

Figure 6: Ablation study of using different KDE kernels and kernel bandwidth.

Figure 7: Ablation study of using different OCSVM kernels and $\nu$ values.

| Noise Type | Gaussian | | Uniform | |
|---|---|---|---|---|
| Noise Level | AUC | Drop | AUC | Drop |
| 0 | 0.949 | - | 0.949 | - |
| 1/255 | 0.929 | -0.020 | 0.934 | -0.015 |
| 2/255 | 0.910 | -0.019 | 0.920 | -0.014 |
| 4/255 | 0.886 | -0.024 | 0.900 | -0.020 |
| 8/255 | 0.867 | -0.019 | 0.880 | -0.020 |
| 16/255 | 0.834 | -0.033 | 0.856 | -0.024 |
| 32/255 | 0.784 | -0.050 | 0.813 | -0.043 |

Table 4: Performance of adversarial anomaly detector on distinguishing noisy benign images and adversarial images.

### D.2  Robustness To Harmless Random Noise

While random noise can be viewed as a perturbation to clean images, they do not generally result in wrong predictions except at high noise levels. A good adversarial anomaly detector should be able to distinguish noisy benign images from adversarial images. To evaluate this behavior we train anomaly detectors on benign images (without noise) and test on noisy benign images and adversarial images. As additive noise, we use either zero-mean Gaussian noise with standard deviation set to a

| Dataset | Method | $l_\infty$ Attacks | | | $l_2$ Attacks | | $l_0$ Attacks | | Combined | Overall |
|---|---|---|---|---|---|---|---|---|---|---|
| | | PGD10 | FGSM | BIM | DeepFool | CW | SparseFool | OnePixel | AutoAttack | |
| CIFAR10 | HEAD+OCSVM (VGG16) | 0.999 | 0.999 | 0.999 | 0.841 | 0.941 | 0.985 | 0.821 | 0.988 | 0.947 |
| | HEAD+KDE (VGG16) | 1.000 | 0.999 | 1.000 | 0.846 | 0.943 | 0.986 | 0.825 | 0.989 | 0.949 |
| | HEAD+OCSVM (ResNet18) | 0.999 | 0.999 | 0.999 | 0.786 | 0.915 | 0.969 | 0.893 | 0.983 | 0.943 |
| | HEAD+KDE (ResNet18) | 1.000 | 1.000 | 0.999 | 0.790 | 0.916 | 0.970 | 0.894 | 0.983 | 0.944 |
| CIFAR100 | HEAD+OCSVM (VGG16) | 0.999 | 0.999 | 0.998 | 0.728 | 0.814 | 0.898 | 0.819 | 0.906 | 0.895 |
| | HEAD+KDE (VGG16) | 0.999 | 1.000 | 0.998 | 0.733 | 0.816 | 0.901 | 0.820 | 0.907 | 0.897 |
| | HEAD+OCSVM (ResNet18) | 0.999 | 0.999 | 0.998 | 0.681 | 0.775 | 0.837 | 0.832 | 0.890 | 0.876 |
| | HEAD+KDE (ResNet18) | 0.999 | 0.999 | 0.998 | 0.676 | 0.772 | 0.838 | 0.831 | 0.889 | 0.875 |
| SVHN | HEAD+OCSVM (VGG16) | 1.000 | 1.000 | 1.000 | 0.868 | 0.975 | 0.992 | 0.954 | 0.994 | 0.973 |
| | HEAD+KDE (VGG16) | 1.000 | 1.000 | 1.000 | 0.917 | 0.979 | 0.994 | 0.946 | 0.994 | 0.979 |
| | HEAD+OCSVM (ResNet18) | 1.000 | 1.000 | 1.000 | 0.811 | 0.955 | 0.989 | 0.958 | 0.990 | 0.963 |
| | HEAD+KDE (ResNet18) | 1.000 | 1.000 | 1.000 | 0.815 | 0.956 | 0.989 | 0.958 | 0.990 | 0.964 |

Table 5: The ROC AUC performance on detecting eight adversarial attacks using VGG16 and ResNet18.

specified noise level, or zero-mean uniform noise with maximum value equal to a specified noise level. Table 4 details overall performance under six different noise levels using the KDE detector. The gray band in the table represents the noise level equivalent to the perturbation budget used in the adversarial attacks. We observe that when noise levels are low, the performance of the detectors does not drop significantly, and remains higher than 85% AUC. Even when the noise level is double that of the adversarial perturbation budget (*i.e.*, noise level=16/255), the performance is still above 80% AUC. In general, HEAD-based anomaly detectors appear to be robust to random noise no larger than perturbation budgets, while experiencing larger performance drop under strong noise (*e.g.*, noise level=32/255).

### D.3 Generalizability To Other Network Architectures

Other than VGG16 [46], there are many network architectures, *e.g.*, ResNet [17], DenseNet [19], have been widely used and outperform VGG16 in many computer vision tasks. To validate the generalizability of our method, we conduct anomaly detection on another network architecture, ResNet18, using the same evaluation protocol in section 3.2. More specifically, we train ResNet18 models on CIFAR10, CIFAR100 and SVHN dataset, respectively, and use eight attacks to generate corresponding adversarial images. We quantitatively evaluate the HEAD's anomaly detection ability by distinguishing benign images and adversarial images using ROC AUC. We use the same 32-dimensional features for LSCFand 5-dimensional features for HF. The HFare obtained by computing the Hessian $l_1$ modulus of the input and intermediate features after each residual layers. table 5 presents the results of anomaly adversarial detection using VGG16 and ResNet18.

We find that HEAD obtains similar performance on $l_\infty$ and $l_0$ attacks when using ResNet18 compare to VGG16. While on $l_2$ attacks, the ResNet18 based HEADcan have performance drop up to 10.2% (DeepFool attack on SVHN). However, the overall anomaly adversarial detection performance on ResNet18 models still reaches 94.4%, 97.6% and 96.4% on CIFAR10, CIFAR100 and SVHN, respectively, which validates the generalizability of our method on other DNN models.

### D.4 Binary Benign/Attack Classification vs. Anomaly Detection

Anomaly detection, in general, does not require examples of the outliers, *i.e.* the adversarial images in this study. An interesting question is what, if any, performance improvement could be gained by incorporating knowledge of the adversarial examples? To answer this question, we use a binary benign/attack classifier to provide an upper bound on the performance, where we train neural networks on benign and adversarial images as inputs with image class (benign or adversarial) as the output. The binary classifier has the same architecture as the previously described LID and Mahalanobis models in Section 3.3. Figure 8 compares supervised training with anomaly detection using three different input features: LID, Mahanolobis, and HEAD. Across all input features, we find that supervised training provides better performance over the corresponding anomaly detector. However, anomaly detection using HEAD features *of only benign samples* show a much smaller performance gap with supervised

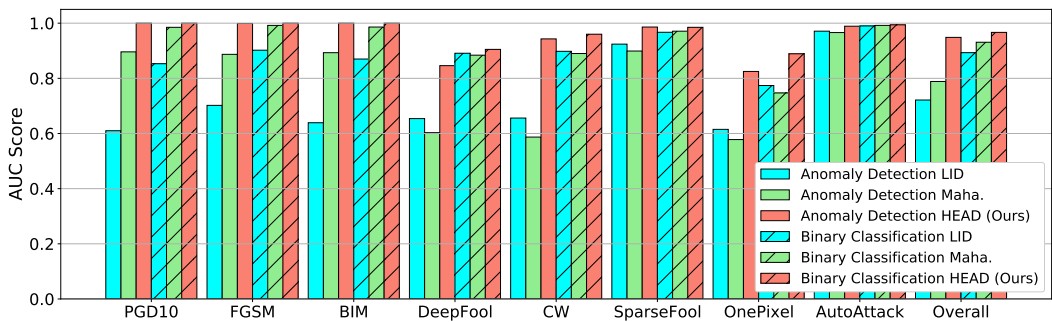

Figure 8: The performance of adversarial detection using anomaly detection and binary classification on CIFAR10. The results of anomaly detection and binary classification are shown in pure color bars and shadowed bars, respectively. The overall performance for binary classification is the average performance of eight attacks.

training, and performance is quite comparable, reinforcing the suitability of HEAD features for adversarial image detection, since knowledge of potential attacks and the ability to generate samples from these attacks is not practical.

