# OpenReview forum: "Attack-Agnostic Adversarial Detection"
_NeurIPS.cc/2022/Workshop/TSRML — TSRML2022_

### Official Review · Reviewer_5vKE · 2022-10-10
**Another static adversarial detection method.**

**Overall Rating:** 6

**Summary:**

The authors submitted another adversarial detection method based on the least significant component and Hessian features. In order to detect unseen attacks, they introduce anomaly detectors. Extensive evaluations are provided. Comparisons to classical detectors like LID or Mahalanobis detectors are given.

**Strengths:**

The paper is clearly written and the main points are easily understandable and explained thoroughly enough.


**Weaknesses:**

 - Significance - In light of insufficient adaptive evaluations [1], it is difficult to understand if this paper contributes meaningfully to the study of adversarial defenses.

[1] https://arxiv.org/pdf/2202.13711.pdf

**Overall Recommendation:**

The authors present an anomaly detection framework for identifying adversarial examples by measuring the statistical deviations caused by adversarial perturbations, which is a novel approach to detecting adversarial examples. There are plenty of evaluations that give valuable insights. There should be made thoughts to adaptive defenses.



**Review Confidence:**

3: The reviewer is fairly confident that the evaluation is correct

---

### Official Review · Reviewer_zcey · 2022-10-19
**Review of paper 'Attack-Agnostic Adversarial Detection'**

**Overall Rating:** 4

**Summary:**

This paper proposes an anomaly detection framework for adversarial detection that measures statistical deviation caused by adversarial perturbations on two proposed features, Least Significant Component Feature (LSCF) and Hessian Feature (HF).

**Strengths:**

The paper is well written.

**Weaknesses:**

1. Although this paper has empirical results to show the diversity between LSCF and HF on benign models and malicious models, they still can not convince me those features have something to do with adversarial attacks because of the lack of logical reasons why those features work.

2. The paper does not explain the connection between those two features. It's totally fine if the results are based on one single feature, but if the provided method is based on more than one feature, I believe the author should explain the connection between those features.

3. The experiment results in section 3 are all based on both those two features, the author should provide the comparison between HEAD and detection based on one single feature.

4. The paper does not show how much data they use to do such an adversarial detection.

**Overall Recommendation:**

Overall, I think this paper lacks the logical reason why they choose those features for the detection. The paper also misses experiments to compare those two features.

**Review Confidence:**

4: The reviewer is confident but not absolutely certain that the evaluation is correct

---

### Official Review · Reviewer_X8HR · 2022-10-20
**Interesting paper but lacks some methodology details**

**Overall Rating:** 6

**Summary:**

Adversarial detection usually requires training the detector with adversarial examples. In this work, the authors proposed a feature extractor for anomaly detection, HEAD, for attack-agnostic adversarial detection. HEAD extracts two features: the Least Significant Component Feature (LSCF) and Hessian Feature (HF), where LSCF measures deviations between clean and adversarial data, and HF measures the smoothness of the loss function’s landscape. The adversarial detection frameworks with HEAD show great performance.

**Strengths:**

The idea is novel: the authors leveraged anomaly detection to do adversarial detection.

The methodology is practical: the authors used matrix decomposition to find the least eigen basis, and found out the least eigenvalues give the most differentiability between clean and adversarial features. Also, utilizing the second-order derivative or Hessian to find out the smoothness is a good idea, since it largely depends on the model.

The performance is promising: Using extracted features by HEAD, the experiment results in Table 1 and Table 2 show great performance and cross-model generalizability in multiple datasets, even compared with binary classification upper bound.

**Weaknesses:**

The pipeline is not clear. To implement HEAD, first, we need to train an adversarial model for HF, like in formulas (4) and (5), generating an adversarial example under a L-p norm. This means that HEAD needs adversarial examples. However, on page 2, line 68, section 2.1, it states that the model trained on only begin examples. LSCF has a similar problem, in page 3, line 95, it says we need ${p’}_i$ to be the mapping of the adversarial perturbed image on the $i$th eigenvector, so LSCF also needs adversarial examples. Basically, we need to use both clean and adversarial examples for LSCF and HF, so we are basically doing a binary classification for clean and adversarial examples in an implicit way.

Moreover, the author didn’t clearly state how to compose LSCF and HF (features), and send them to either OCSVM or KDE anomaly detector.  The authors may consider plotting a figure showing the pipeline.

The storyline is not clear. The authors only consider adversarial attacks that are regular or change few pixels. For example, all the white box attacks are gradient-based attacks, and they are regular in sense of data distribution difference between clean and adversarial data. For black-box attacks, these attacks are changing a few pixels, as few as possible. Though the authors mentioned harmless random noise, the authors may consider using natural out-of-distribution examples. Since the proposed adversarial detection is attack-agnostic, it should work for irregular adversarial examples such as out-of-distribution in the wild.

**Overall Recommendation:**

The idea is novel and the experimental results seem good. However, the organization and the storyline are not very clear.

**Review Confidence:**

4: The reviewer is confident but not absolutely certain that the evaluation is correct

---

### Decision · Program_Chairs · 2022-10-23

Accept